# The Psychological Antecedents to COVID-19 Vaccination among Community Pharmacists in Khartoum State, Sudan

**DOI:** 10.3390/medicina59050817

**Published:** 2023-04-22

**Authors:** Einass M. Satti, Yasir Ahmed Mohammed Elhadi, Kannan O. Ahmed, Alnada Ibrahim, Ahlam Alghamdi, Eman Alotaibi, Bashir A. Yousif

**Affiliations:** 1Department of Pharmacy Practice, Faculty of Pharmacy, University of Khartoum, Khartoum 11111, Sudan; nosasatti0@gmail.com; 2Department of Public Health, Sudanese Medical Research Association, Khartoum 11111, Sudan; hiph.yelhadi@alexu.edu.eg; 3Department of Clinical Pharmacy and Pharmacy Practice, Faculty of Pharmacy, University of Gezira, Wad Medani 21112, Sudan; omerkannan@gmail.com; 4Department of Pharmacy Practice, College of Pharmacy, Princess Nourah Bint Abdulrahman University, P.O. Box 84428, Riyadh 11671, Saudi Arabia; ahaalghamdi@pnu.edu.sa (A.A.); emelotaibi@pnu.edu.sa (E.A.); 5Department of Pharmacology, Faculty of Pharmacy, University of Khartoum, Khartoum 11111, Sudan

**Keywords:** community pharmacy, COVID-19 vaccine, the 5C model, psychological determinants

## Abstract

*Background and Objectives*: Little is known regarding the 5C psychological antecedents to COVID-19 vaccination among pharmacists in low- and middle-income countries. This study aimed to assess the acceptance of COVID-19 vaccination and its psychological antecedents among community pharmacists in Khartoum State, Sudan. *Materials and Methods*: A cross-sectional study was conducted from July to September 2022. A self-administered questionnaire was used to collect data about sociodemographic and health status characteristics, vaccine acceptance, and the 5C psychological antecedents to vaccination. Stepwise logistic regression analysis was conducted, and results were presented using odds ratios (ORs) and their corresponding 95% confidence intervals (CIs). *Results*: A total of 382 community pharmacists participated in the current study, with a mean age of 30.4 ± 5.6 years. Nearly two-thirds of the participants (65.4%) were females, and the majority (74.9%) have received or intended to receive the COVID-19 vaccination. Vaccine acceptance was significantly associated with the following psychological antecedents to vaccination: confidence, complacency, constraints, and calculation (*p* < 0.001). Results of the logistic regression showed that confidence in vaccines [OR = 6.82 (95% CI = 3.14–14.80)], conspiracy beliefs [OR = 0.44 (95% CI = 0.23–0.85)], and constraints to vaccination [OR = 0.18 (95% CI = 0.06–0.56)] were the significant determinants of vaccine acceptance. *Conclusion*: The study revealed important predictors of COVID-19 vaccine acceptance that can be used to guide policymakers in designing target-oriented interventions that can improve the vaccine acceptance rate among community pharmacists in Sudan. These findings suggest that interventions to promote vaccine acceptance among pharmacists should focus on building confidence in vaccines and providing accurate information about the safety and efficacy of the COVID-19 vaccine, and reducing constraints to vaccination.

## 1. Introduction

Since the first COVID-19-positive case was confirmed in Sudan on 13 March 2020, the number of confirmed cases and associated deaths has grown beyond the country’s ability to respond [1]. In Sudan, up to 21 March 2023, there have been 63,899 confirmed cases of COVID-19 with 5026 deaths reported by the World Health Organization (WHO) [2]. Sudan received the first COVID-19 vaccine from AstraZeneca via the COVID-19 Vaccines Global Access (COVAX) facility [3]. Up to 4 March 2023, a total of 18,495,743 vaccine doses have been administered in Sudan, with 10,576,238 individuals fully vaccinated, 12,628,561 having received one dose of the vaccine, and 2,765,574 having received a booster dose [2].

Vaccine hesitancy, defined as the reluctance or refusal to vaccinate despite the availability of vaccines, has been present for as long as vaccines have been available [4]. Vaccine hesitancy is a complex issue that has evolved over time and is influenced by a variety of social, cultural, and political factors [4]. In the early days of vaccines, vaccine hesitancy was driven primarily by concerns about safety and efficacy. As vaccine technology improved and the benefits of vaccination became more apparent, vaccine hesitancy became less common. However, in recent years, vaccine hesitancy has been on the rise again, fueled in part by misinformation and mistrust of the medical establishment [5,6]. The internet and social media have played a significant role in the spread of vaccine misinformation [7]. False claims about vaccines causing autism, infertility, or other health problems have circulated widely, despite being debunked by numerous studies [8,9]. In addition, some people have expressed concerns about the speed with which COVID-19 vaccines were developed, despite the rigorous testing and safety measures that were followed to prove vaccine efficacy and effectiveness [10,11]. The COVID-19 pandemic has also led to increased polarization and politicization of vaccine hesitancy [12,13,14]. Some people have refused to get vaccinated because of their political beliefs or because they see vaccination as a symbol of government overreach [15,16]. Emerging research suggested that vaccination hesitancy has increased across many populations, even among healthcare workers [17,18].

Various models have been conceptualized to explain behavioral and physiological determinants of vaccine acceptance [19,20,21]. The 5C model of psychological antecedents to vaccination is a framework that was developed to explain the various factors that influence an individual’s decision to get vaccinated [22]. Since the development of the 5C model, researchers have used it to study vaccine hesitancy and uptake in various populations and contexts. The model has been adapted and modified in different ways to fit specific research questions and has explained a greater extent of vaccination variance than other existing models, although it has not been tested alongside the Health Belief Model and the Theory of Planned Behavior [22]. Each antecedent of the 5C model describes individual preferences or psychological and mental representations of the environment in the respondent’s life [23]. Therefore, inequality or political realities that deteriorate confidence in the health system could become the driver of insufficient confidence for a person in one country [22].

Previous studies on vaccine hesitancy in Sudan showed that pharmacists reported low levels of COVID-19 vaccine acceptance [17,24]. A cross-sectional study conducted over the period 15–22 December 2020 in Greece showed that more than one-third of pharmacists did not receive the COVID-19 vaccine [25]. Moreover, a study conducted in Italy during the period December 2020 to February 2021 reported that 36% were unwilling to receive the COVID-19 vaccine [26]. In Pakistan, a qualitative study conducted during 2021 showed that factors leading to COVID-19 vaccine hesitancy were adverse reactions, the cost of the COVID-19 vaccine, and limited data on the safety and efficacy profile of the COVID-19 vaccine [27].

According to prior studies, vaccine hesitancy is primarily the result of an individual’s decision-making process, which is influenced by that person’s sentiments toward vaccination in general or a specific vaccine, as well as both enablers and obstacles to immunization [6,15]. Thus, it is essential to understand which psychological drivers determine whether to refuse or delay the vaccination so that targeted interventions can be developed to reduce vaccine hesitancy and increase vaccine demand [28,29]. This study aimed to assess the acceptance of COVID-19 vaccination and its psychological antecedents among community pharmacists in Khartoum State, Sudan.

## 2. Materials and Methods

### 2.1. Study Design, Setting, and Target Population

This was a cross-sectional study reported using the Strengthening the Reporting of Observational Studies in Epidemiology (STROBE) statement [30]. This study was carried out among community pharmacists in Khartoum State in Sudan. All registered community pharmacists in Khartoum State who worked in community pharmacies during the period of data collection and who agreed to participate in the study were included.

### 2.2. Sampling

The sample size was calculated using the Epi info-7 software (Epi-Info 7 for Dos version 3.5.1 software, Centers for Disease Control and Prevention, Clifton Road Atlanta, USA), based on the total number of Sudanese community pharmacists in Khartoum State accounting for 7128 community pharmacists as per the latest statistical pharmaceutical report of the Federal Ministry of Health in Sudan. Assuming that 50% is the expected proportion of vaccine acceptance, using an alpha error of 0.05, and a margin of error of 5%, the minimum required sample size was 364 community pharmacists. The sample size was increased by 10% to account for sampling bias and the need for stratification. Participants were recruited through a consecutive sampling technique (a non-probability sampling technique where the first subject that meets the inclusion criteria will be selected for the study).

### 2.3. Data Collection

The data were collected using a validated, structed, and self-administered questionnaire that consisted of two sections. The first section contains the socio-demographic and health status characteristics (age, sex, residence, marital status, presence of comorbidities, previous infection with COVID-19, belief regarding the origin of the virus, preference for COVID-19 vaccine based on developer, main source of information about COVID-19 vaccination) and COVID-19 vaccine acceptance. The second part comprises the validated version of the 5C scale for psychological antecedents to vaccination. It contains 15 questions covering the 5C domains that were selected based on a previous protocol by Betsch et al. [31]: confidence, complacency, constraints, calculation, and collective responsibility, each of which had 3 questions to be answered using a 7-point Likert scale (1 = strongly disagree to 7 = strongly agree). The cutoff point for the confidence, complacency, constraints, calculation, and collective responsibility domains were 5.7, 4.7, 6.0, 6.3, and 6.2, respectively. Each of the 5 sub-scales (confidence, complacency, constraints, calculation, and collective responsibility) was assessed by 3 rating items on a 7-point scale (1 = strongly disagree, 2 = moderately disagree, 3 = slightly disagree, 4 = neutral, 5 = slightly agree, 6 = moderately agree, 7 = strongly agree). The mean scores of items under each subscale were computed, with a higher mean score indicating stronger agreement with the corresponding sub-scale. Using the 5C scale does not lead to a total score providing a sample’s absolute state of hesitancy. Rather, it allows for a valid assessment of the different psychological antecedents. The data were collected from 1 July to 30 September 2022.

### 2.4. Operational Definitions of the Measured 5C Variables in the Study [22,32]

**Confidence:** “Trust in the vaccine’s effectiveness and safety, the system that delivers them (including the competence and reliability of the health services and health specialists), and the motivations of policymakers who decide on the need for vaccines”.

**Complacency:** “It exists where perceived risks of vaccine-preventable infections are descending, and vaccination is not considered a necessary preventive step”.

**Constraints:** “Are issues when physical availability, willingness-to-pay, affordability, geographical accessibility, the appeal of immunization service affect uptake and ability to understand (language and health literacy)”.

**Calculation:** “Refers to individuals’ engagement in comprehensive information searching”.

**Collective responsibility**: “The willingness to protect other people by one’s vaccination using herd immunity”.

### 2.5. Data Analysis

The data were analyzed using International Business Machines [IBM]. Statistical Package for Social Sciences [SPSS] for Windows, Version 23.0 software [Armonk, NY: IBM Corp]. The quantitative variables were expressed using mean ± SD, whereas counts (%) were utilized to describe the categorical variables. The chi-squared test was used to estimate the factors associated with vaccine acceptance. The respondents were categorized as (Yes/No) based on their mean percentage scores on the 5C scale. A stepwise binary logistic regression model using all variables was conducted to estimate the significant predictors for confidence, complacency, calculation, constraints, and collective responsibility. Odds ratios (ORs) and 95% confidence intervals (CIs) are reported, and *p* < 0.05 was considered statistically significant.

### 2.6. Ethical Consideration

Ethical approval (FPEC-13-2022, 8 June 2022) was obtained from the Ethical Committee at the Faculty of Pharmacy, University of Khartoum. Informed consent was obtained from the participants ahead of administering the questionnaire. The participants were informed about the importance and objectives of the study. The right to withdraw at any time was clearly stated. To ensure confidentiality, the responses were coded using numbers rather than personal identifiers.

## 3. Results

### 3.1. Respondents’ Profile and Vaccine Acceptance

A total number of 382 community pharmacists were surveyed for vaccine acceptance in the current study. The mean age of the study population was 30.4 ± 5.6 years, with more than half being females (65.4%) and single (62.6%). More than one-third (34.6%) of the study self-reported their monthly income within the median quantile (81,000 to 100,000 Sudanese pounds). Regarding health status, nearly one-fifth (19.1%) of participants reported suffering from at least one chronic disease (hypertension, diabetes, or asthma), and nearly two-thirds (63.1%) of participants were clinically diagnosed with COVID-19. Half (50.3%) of the studied population believed the origin of the current pandemic was a natural source (animals), while nearly one-third (29.1%) of participants reported conspiracy beliefs. Among those vaccinated or intending to become vaccinated, the most preferred type of COVID-19 vaccine was the Pfizer-BioNTech vaccine (50.3%), followed by the Oxford-AstraZeneca vaccine (32.7%) Many participants (43.5%) reported seeking information about COVID-19 vaccination primarily from scientists/scientific journals (Table 1).

### 3.2. The 5C Psychological Antecedents to Vaccination

Illustrated in Figure 1 are the levels of the 5C psychological antecedents to COVID-19 vaccination among participants. More than half of the respondents (53.9%) had no confidence in COVID-19 vaccines, while only 12.8% had complacency, and (8.1%) reported constraints to vaccination, 47.1% were involved in the calculation domain, and only 8.4% had collective responsibility.

### 3.3. Factors Associated with Vaccine Acceptance

Vaccine acceptance was significantly associated with infection status regarding COVID-19 (*p* = 0.036). Moreover, there was a statistically significant difference in the level of vaccine acceptance related to participants’ monthly income (*p* = 0.005). Furthermore, there was a statistically significant difference in vaccine acceptance among participants related to different beliefs regarding the origin of the virus (*p* < 0.001), information sources (*p* < 0.001), and vaccine preference (*p* < 0.001). However, there was no significant association noted between vaccine acceptance and participants’ age, gender, marital status, or chronic diseases status (Table 2).

### 3.4. The 5Cs Psychological Antecedents and Vaccine Acceptance

As illustrated in Table 3, there was a statistically significant association between vaccine acceptance and the levels of confidence, complacency, constraints, and calculation (*p* < 0.001). However, there was no significant difference in vaccine acceptance related to participants’ collective responsibility (*p* = 0.404).

### 3.5. Determinants of COVID-19 Vaccine Acceptance among Participants

The results of stepwise logistic regression analysis for the predictors of COVID-19 vaccine acceptance are shown in Table 4. Stepwise regression for the nine significant factors associated with vaccine acceptance through univariate analysis resulted in a model that included three variables: beliefs about the origin of the virus, confidence in vaccines, and constraints to vaccination. Conspiracy beliefs were associated with lower odds of accepting COVID-19 vaccines [OR = 0.44 (95% CI = 0.23–0.85)]. Regarding the 5C model, participants with confidence in COVID-19 vaccines were more likely by six times to accept the vaccine [OR = 6.82 (95% CI = 3.14–14.80)]. Moreover, participants who faced constraints to vaccination were less likely to receive the vaccine [OR = 0.18 (95% CI = 0.06–0.56)]. These variables explained 37% of the variance in vaccine acceptance.

## 4. Discussion

Assessing the psychological antecedents of vaccination is crucial because it helps researchers and public health authorities understand vaccine hesitancy, design effective interventions, tailor communication messages, and identify at-risk populations. By understanding the psychological factors that influence vaccine decision-making, targeted public health campaigns can be developed to increase vaccination rates, reduce the spread of diseases, and ultimately improve public health outcomes. To our knowledge, this is the first study to assess the psychological antecedents to vaccination in a population of community pharmacists in resource-restricted settings.

In this cross-sectional study, 74.9% of pharmacists received or intended to receive the COVID-19 vaccination. While this rate may be relatively high in comparison to the general population, a 74% acceptance rate may be considered somewhat low compared to other healthcare professions [33,34]. Healthcare workers, including pharmacists, have a crucial role to play in promoting vaccination and protecting patients and themselves from vaccine-preventable diseases [35,36]. As such, higher vaccine acceptance rates among healthcare workers are generally preferred to minimize the risk of disease transmission within healthcare settings. Compared to other countries, the Kuwaiti survey conducted in 2021 reported a higher proportion of vaccine acceptance, where 80.1% of pharmacists said they intended to receive COVID-19 vaccination [37]. On the other hand, several cross-sectional studies conducted in Greece (2020) [25], Poland (2020) [38], Italy (2020–2021) [26], Nigeria (2021) [39], and Zambia (2021) [40] documented that pharmacists and pharmacy students demonstrated a low willingness to accept COVID-19 vaccination.

This study examined the 5C psychological factors that influence vaccination decisions among pharmacists. The results showed that most pharmacists had high levels of calculation and confidence in their decision to vaccinate. Only a small percentage of pharmacists reported complacency (12.8%), constraints to vaccination (8.1%), and collective responsibility (8.4%). The levels of confidence, complacency, constraints, and calculation were associated with pharmacists’ vaccine acceptance. In the same way, a cross-sectional study performed in Italy reported that the factors significantly associated with a willingness to receive the COVID-19 vaccination were confidence in the capacity of vaccines to prevent vaccine-preventable diseases, fear of contracting SARS-CoV-2 infection, and considering COVID-19 vaccination to be the best strategy to counteract the SARS-CoV-2 infection [26]. On the other hand, findings obtained in Kuwait reported high levels of collective responsibility and confidence and lower levels of constraints and calculation [37].

This study reveals that conspiracy beliefs were associated with lower odds of accepting COVID-19 vaccines. In support of this finding, many studies showed that increased belief in vaccine conspiracy theories is a main predictor of vaccination hesitancy including among healthcare workers [41,42,43,44,45]. Moreover, cross-sectional studies conducted in Greece [25], Zambia [40], and the Eastern Mediterranean Region [17] documented that the potential predictors of vaccine acceptance included being concerned about the side effects of vaccines and suspicion about vaccine effectiveness in preventing COVID-19. Furthermore, participants with confidence in COVID-19 vaccines were more likely by six times to accept the vaccine. Moreover, participants who faced constraints to vaccination were less likely to receive the vaccine. Also, Sudanese cross-sectional studies reported consistent results, where lower acceptability of the COVID-19 vaccine was significantly associated with increased perceived harm from the vaccine and lack of confidence in the source of the vaccine, organizations, and government supervising the COVID-19 vaccination process [24,46]. Also in Nigeria, Pakistan, Italy, and the United States, many cross-sectional studies reported that fear of adverse effects was expressed as the main reason for vaccination hesitation [26,27,39,47]. Facing constraints to vaccination was a significant determinant of vaccination intention among the current study participants. Ensuring pharmacists have access to COVID-19 vaccination is essential for protecting their health, promoting vaccination, and maintaining healthcare services during the pandemic [48,49].

### Limitations

This study encountered the following limitations: the use of nonprobability sampling introduces the risk of sampling bias, which limits the generalization of study findings. Further, this study included only community pharmacists working in Khartoum State, while pharmacists working in other settings such as hospitals were excluded; these categories cannot be ignored, and larger studies with different contexts may yield different outcomes. Moreover, the study was conducted only in Khartoum, and this would limit the generalization of the study results to community pharmacists in other Sudanese states. Also, this study was based on recalling past events; hence the results are prone to misclassification (recall) bias. In addition, the current study used a self-reported tool to assess vaccine acceptance among community pharmacists in Sudan. Therefore, the findings of the study might have been affected by social desirability bias. Finally, the study has not assessed acceptance of the COVID-19 vaccine booster dose among pharmacists in Sudan.

## 5. Conclusions

In conclusion, the study found that 74.9% of pharmacists in Khartoum State had accepted the COVID-19 vaccine. This is a relatively high acceptance rate, indicating that pharmacists are willing to take steps to protect themselves and their patients from COVID-19. However, much still needs to be done in order to increase vaccine acceptance among this population. The study also revealed that pharmacists who reported higher levels of calculation and confidence were more likely to accept the vaccine. The study revealed important predictors of COVID-19 vaccine acceptance that can be used to guide policymakers in designing target-oriented interventions that can improve the vaccine acceptance rate among community pharmacists in Sudan. These findings suggest that interventions to promote vaccine acceptance among pharmacists should focus on building confidence, providing accurate information about the safety and efficacy of the COVID-19 vaccine, and reducing constraints to vaccination.

## Figures and Tables

**Figure 1 medicina-59-00817-f001:**
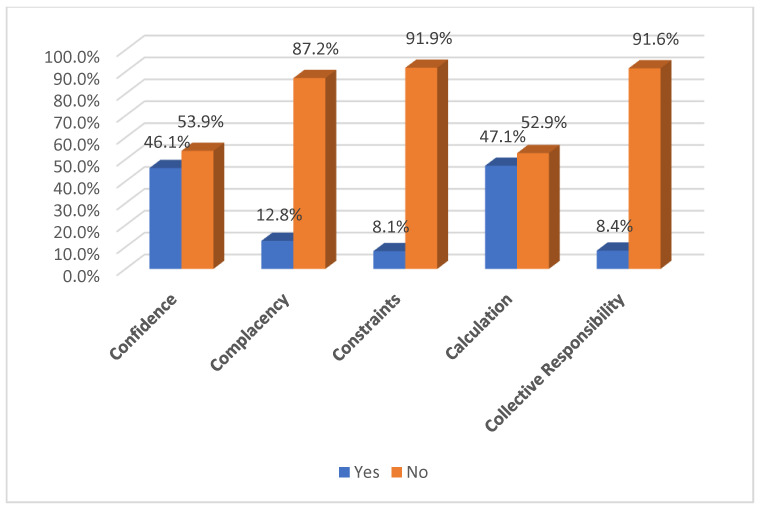
The 5C psychological antecedents to COVID-19 vaccination among community pharmacists in Khartoum state, Sudan (*n* = 382).

**Table 1 medicina-59-00817-t001:** Participants’ socio-economic characteristics, health status, and perception towards COVID-19 vaccination (*n* = 382).

Variables	Categories	*n*	(%)
**Age (years)**	20–25	68	(17.8)
	26–35	254	(66.5)
	36–45	54	(14.1)
	Above 45	6	(1.6)
**Gender**	Female	250	(65.4)
	Male	132	(34.6)
**Marital status**	Single	239	(62.6)
	Married	129	(33.8)
	Divorced or widowed	14	(3.7)
**Monthly income (Sudanese pounds)**	Below 61,000	89	(23.3)
	61,000–80,000	97	(25.4)
	81,000–100,000	132	(34.6)
	Above 100,000	64	(16.8)
**Chronic diseases**	Yes	73	(19.1)
	No	309	(80.9)
**Infection with COVID-19**	Yes	241	(63.1)
	No	141	(36.9)
**Origin of SARS-CoV-2**	Natural sources from animals	192	(50.3)
	Man-made virus and part of a conspiracy plan	111	(29.1)
	No opinion	79	(20.7)
**Received or intend to receive COVID-19 vaccination**	Yes	286	(74.9)
	No	96	(25.1)
**Vaccine preference**	Pfizer-BioNTech COVID-19 vaccine.	192	(50.3)
	Oxford-AstraZeneca COVID-19 vaccine	125	(32.7)
	Johnson and Johnson COVID-19 vaccine.	45	(11.8)
	Sinopharm COVID-19 vaccine	20	(5.2)
**The main source of information about COVID-19 vaccination**	Scientists/scientific journals.	166	(43.5)
	Social media platforms	94	(24.6)
	TV programs, newspapers, and news releases	71	(18.6)
	Doctor/health care workers	44	(11.5)
	Others	7	(1.8)

**Table 2 medicina-59-00817-t002:** Factors associated with COVID-19 vaccine acceptance among community pharmacists in Khartoum state, Sudan (*n* = 382).

Socio-Economic and Health Status Characteristics	Categories	Vaccine Acceptance				Chi-Squared	*p*-Value
		Yes (*n* = 286)		No (*n* = 96)			
		*n*%		*n*%			
**Age (years)**	20–25	45	66.2%	23	33.8%	5.156	0.164
	26–35	193	76.0%	61	24.0%		
	36–45	42	77.8%	12	22.2%		
	Above 45	6	100.0%	0	0%		
**Gender**	Female	183	73.2%	67	26.8%	1.071	0.301
	Male	103	78.0%	29	22.0%		
**Marital status**	Single	176	73.6%	63	26.4%	1.830	0.423
	Married	101	78.3%	28	21.7%		
	Divorced or widowed	9	64.3%	5	35.7%		
**Monthly income (SDG)**	Below 61,000	55	61.8%	34	38.2%	13.062	0.005 *
	61,000–80,000	71	73.2%	26	26.8%		
	81,000–100,000	107	81.1%	25	18.9%		
	Above 100,000	53	82.8%	11	17.2%		
**Chronic disease**	Yes	56	76.7%	17	23.3%	0.163	0.686
	No	230	74.4%	79	25.6%		
**Infection with COVID-19**	Yes	189	78.4%	52	21.6%	4.383	0.036 *
	No	97	68.8%	44	31.2%		
**Origin of SARS-CoV-2**	Natural sources from animals	168	87.5%	24	12.5%	32.732	<0.001 *
	Man-made virus and part of a conspiracy plot	69	62.2%	42	37.8%		
	No opinion	49	62.0%	30	38.0%		
**Vaccine preference**	Pfizer-BioNTech COVID-19 vaccine.	150	78.1%	42	21.9%	12.670	<0.001 *
	Oxford-AstraZeneca COVID-19 vaccine	97	77.6%	28	22.4%		
	Johnson and Johnson COVID-19 vaccine.	24	53.3%	21	46.7%		
	Sinopharm COVID-19 vaccine	15	75.0%	5	25.0%		
**The main source of information about COVID-19 vaccination**	Scientists/scientific journals.	149	89.8%	17	10.2%	38.518	<0.001 *
	Social media platforms	54	57.4%	40	42.6%		
	TV programs, newspapers, and news releases	50	70.4%	21	29.6%		
	Doctor/health care workers	29	65.9%	15	34.1%		
	Others	4	57.1%	3	42.9%		

SDG: Sudanese pounds. *: Statistically significant at *p* ≤ 0.05.

**Table 3 medicina-59-00817-t003:** Association between the 5Cs psychological antecedents and vaccine acceptance among community pharmacists in Khartoum State (*n* = 382).

The 5Cs Domains	Responses	Vaccine Acceptance				Chi-Squared	*p*-Value
		Yes (*n* = 286)		No (*n* = 96)			
		*n*%		*n*%			
**Confidence**	Yes	161	91.5%	15	8.5%	47.846	<0.001 *
	No	125	60.7%	81	39.3%		
**Complacency**	Yes	26	53.1%	23	46.9%	14.208	<0.001 *
	No	260	78.1%	73	21.9%		
**Constraints**	Yes	14	45.2%	17	54.8%	15.825	<0.001 *
	No	272	77.5%	79	22.5%		
**Calculation**	Yes	153	85.0%	27	15.0%	18.568	<0.001 *
	No	133	65.8%	69	34.2%		
**Collective responsibility**	Yes	22	68.8%	10	31.3%	0.695	0.404
	No	264	75.4%	86	24.6%		

*: Statistically significant at *p* ≤ 0.05.

**Table 4 medicina-59-00817-t004:** Logistic regression analysis for the determinants of COVID-19 vaccine acceptance among community pharmacists in Khartoum state, Sudan.

Predictors	Odds Ratio	95% CI of Odds Ratio		*p*-Value
		Lower	Upper	
**Beliefs about the origin of COVID-19**				
**Natural source from animals^®^**	1.00			
**Man-made virus and part of a conspiracy plot**	0.44	0.23	0.85	0.014
**No opinion**	0.495	0.24	1.01	0.054
**Confidence**				
**Yes**	6.82	3.14	14.80	0.001
**No^®^**	1.00			
**Constraints**				
**Yes**	0.18	0.06	0.56	0.003
**No^®^**	1.00			
**Constant**	2.827			0.019

CI: Confidence interval^®^: reference group.

## Data Availability

Data and other supporting material are available from the corresponding author upon reasonable request.

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
