# Peer review of "The Psychological Antecedents to COVID-19 Vaccination among Community Pharmacists in Khartoum State, Sudan"

_medicina, 2023, doi:10.3390/medicina59050817_

Round 1

Reviewer 1 Report

The topic is very important because it is related to COVID-19 vaccination. I have some minor comments.

1.         (Abstract) p<000 is an unusual expression. Please modify it.

2.         (Abstract) When describing an association of factors with vaccination, you should add “positive” or “negative” for each description.

3.         (Introduction) I understood that there are some previous studies that investigated vaccine hesitancy among pharmacists in other countries, but why is focusing on pharmacists important? Are there any merits in restricting the subjects to pharmacists?

4.         (2.5 data analysis) Could you write all the variables used in the logistic regression analysis?

5.         (Table 3) I think that chi-squared value is not generally shorted as C2.

6.         (Table 4) Didn’t you include other characteristics, such as sex, income, and age in the model? If so, why didn’t you include them?

7.         (Table 4) Positions of p-value and 95%CI should be exchanged.

Reviewer 2 Report

Abstract: the English language is not used having a full understanding of the terms properties- see rows 17-32. Moreover, the normal and scientific structure of the abstract is not respected, as conclusions are missing in the abstract. There is also a short formulation of the psychological model, such as “5C” that is not explained before introducing the notion and should not be used in this situation. 

Materials and method- a very innapropiate description, with references to methods? Never heard. Please, see rows 84-85. The English language has to be completely reformulated. I cannot understand the form and the content of the authors ideas.

Rows 90-143 are formulated by a statistician, but the data seems to be communicated in such a form that can fit into the numbers.

The ethical consideration: again, English is not used properly in this phrases. Please see rows 144-150.

References- there are very few number of references, since this is an original article and requires a much profound understanding of the literature and studies that approached the same matter.

Reviewer 3 Report

The psychological antecedents to COVID-19 vaccination among community pharmacists in Khartoum State, Sudan

Dear authors:

Thank you for this interesting manuscript which will help in the understanding of the phenomenon of vaccine hesitancy:

After reviewing, I found the results interesting. However, the manuscript lack of a certain mastery in this field and a certain scientific rigor especially in its redaction and results interpretation. It needs to be improved. In addition to the used references that need to be updated, the authors tend to compare their results obtained in 2022 with the results of other conducted in 2020 or 2021 where some of them were conducted before the introduction of the vaccine. In this way, one of the most important parameter missed in this study is the evolution of the vaccine hesitancy during time. The particularity of the results related to the COVID-19 pandemic in general and COVID-19 vaccines particularly (contrarily to other diseases and their vaccines) is the rapid evolution in each parameter related to the diseases. Consequently, in the presence of a panel of data regarding vaccine hesitancy in the world, the approach should be based on the description of the evolution in the world and in the country of the study (comparing results of your study with the results of a study conducted in Zambia in 2020 has no sense).

The authors should so, update their references and revise their way of explanation and comparison of their results.

I have also other "minor" remarks:  

Abstract

You should add the period of the study.

You should associate your explanation about the associated factors and determinants (also in the main text). What is the difference between these two parameters? Should you not base on the determinants since it is the final step of the analysis?

You should add small conclusion to the abstract

Introduction

The introduction should be revised and updated with recent references.

Line 39: ….with 4,960 deaths, reported to WHO  (correct as: reported be the WHO).

Line40: What is your objective by including data of May and September ?? You should relate he  data to better understand the evolution. You should also add the most recent data regarding COVID-19 and its vaccines in the world and in your country.

Line 43: ….. received at least a single dose of the vaccine (this may be: received one dose of the vaccines. When you say at least that means that they could receive 2 or 3 doses..).

Line 62: previous studies….(you should add other references since you used "studies").

In this paragraph (62-72) particularly you should describe the evolution of the phenomenon of vaccine  hesitancy in the world and in your country and do not limit your research to "some cases".

Line 70: Greek should be Greece ( correct in all the manuscript).

Line 75-78: According to studies… (add the references of these studies).

At last, you should add the objectives of the study

Materials and methods

Line 99: what are the inclusion and the exclusion criteria?

Results

what are the type and the number of doses of the vaccine that the participants received?

Table 1: vaccine preference: the number should be 286 (those who were vaccinated or intend to get the vaccine). Those who were against vaccination should not have a preference to a vaccine.

You should also improve the quality of the tables. Use black lines and  correct the line yes in " Received or intend to receive COVID-19 vaccination" to be in one line.

Delete the title of figure 1  in the background

Line 181: replace moreover by another word since it is used in line 180.

Line 185: "….and participants’ age, gender, marital status, or suffering from chronic diseases (Table 2)". Correct suffering from chronic diseases to be in accordance with the beginning of the sentence.

Line 201: there were 9 significant factors not 8.

Line 209: "….These variables explained 37% of the variance in vaccine acceptance". What do you mean here exactly? Explain please

Discussion:

All my first "main" remarks (mentioned above) apply to discussion that should be carefully revised and updated. The discussion needs an extensive revision. Additionally, it is too long and the fact that you discuss all your results including those obtained with univariate and logistic regression make the comprehension of your ideas difficult ad reader is lost and could not find the most important results.

Line 256: "….having high efficacy [22]". You should support with other real studies after the introduction of the vaccines.

Line 257: reorder the references or include them just after the country.

Line 261: "…..Sudan was the Oxford- AstraZeneca vaccine". Reformulate to delete the repetition (delete the Oxford- AstraZeneca vaccine).

Lines 262-278: You should explain how did these sources affects the phenomenon of vaccine hesitancy/acceptance.

Line 295-298: "Regarding the 5C model, …… variance in vaccine acceptance". You repeated here the same paragraph of the results. Reformulate.

At last you should add the conclusion of the work.

Reference 18 is incomplete

Round 2

Reviewer 2 Report

Recommendation for a  better structure of the introduction, with 3-4 paragraphs, including: description (importance) of the problem(soundness); proposed work in the study; how this work will increase the soundness of the problem.

Author Response

Response: Thank you for this comment. The introduction of the work was revised and restructured to fit the provided suggestions. Changes were highlighted within the revised manuscript.

Reviewer 3 Report

I would thank the authors for their efforts to improve the quality of the manuscript. However, I Still have multiple concerns about some points:

1. I will begin with your response here:

Table 1: vaccine preference: the number should be 286 (those who were vaccinated or intend to get the vaccine). Those who were against vaccination should not have a preference to a vaccine.

Response: We disagree with this comment. Participants in the pre-contemplation stage and who have not decided yet to receive the vaccine may have vaccine preference towards a specific brand of COVID-19 vaccine. Actually, a preference for a specific vaccine that was not available or runout of stock in the country may be one of the primary reasons to not vaccinate.

Your response here could affect all your results and analysis. In fact, considering hesitancy because the vaccine is not available in the country (or to the end of stock) is not "very appropriate". Also, you do not precise if the hesitant (or reluctant) had definitely or momentarily decided to not get the vaccine and what was their reason for hesitancy.

2. I still have also a concern about your way of writing and you did not take my comments in consideration. In fact, You added just some paragraphs without trying to adapt them with the existing paragraphs. Additionally, You did not answered to my main important comment about the evolution of vaccine hesitancy during time. You should precise that the vaccine hesitancy has changed when compared to the first months of COVID-19 vaccine introduction. Thus you should precise the period of each cited study to better understand the evolution among Pharmacist through the world.

It is also to mention, the fact that you used 50 references did not mean that your study is well referenced. The most important is the quality of the used references.

3. Your discussion should be adapted to the results of the logistic regression

" Response: The difference is based on the statistical adjustment for confounding variables (Adjusted Vs unadjusted analyses). We first conducted a bi-variate analysis to portray significant factors associated with vaccine acceptance before adjustment in order to select variables to be included in the iterations of the final stepwise logistic regression model. The regression model was built based on the following assumption:  [to include  only those variables with less than 0.05 level of Alpha Error in the bi-variate analysis]. "

I understand this but you should update your discussion according to the results of the logistic regression. We cannot treat these two types of analysis separately since the logistic regression complete the bivarite analysis.

I have also other remarks:

Line 18-19: "Little is known… low-and-middle income countries." This statement is not currently correct. Multiple studies are currently available.

Your abstract should be based on the results of the logistic regression.

Line 54-71: You should precise when you are talking about VH in general and when you are talking about COVID-19 VH.

Line 89-90: "Previous….."

a- This statement is not correct for the two studies

b- why did you not use these results to calculate the sample size.

Line 90-97: Here you repeated the same paragraphs without taking my comments into consideration.

Line 99-101 " It is critical …. community" and lines 104-106: " Thus, it is essential …. demand [29,30]". Here you repeated nearly the same statement. Associate and reformulate please

Discussion is the weakest part of the manuscript:

Your discussion should be based on the results of the logistic regression

Some of the added paragraphs are not well placed in the discussion.

Line 246: Replace to our search by another expression.

Line 250-252: While …..other healthcare profession". From where did you take this statement ???

Line 262-282: adapt please with the results of the logistic regression

In Line 283 you discuss the 5C psychological factors, In line 295 you discuss the conspiracy belief and in line 301 you return to the 5C.

Reorder please.
